# Field-linked resonances of polar molecules

Xing-Yan Chen[1,2,5], Andreas Schindewolf[1,2,5], Sebastian Eppelt[1,2], Roman Bause[1,2], Marcel Duda[1,2], Shrestha Biswas[1,2], Tijs Karman[3], Timon Hilker[1,2], Immanuel Bloch[1,2,4] & Xin-Yu Luo[1,2✉]

Scattering resonances are an essential tool for controlling the interactions of ultracold atoms and molecules. However, conventional Feshbach scattering resonances[1], which have been extensively studied in various platforms[1–7], are not expected to exist in most ultracold polar molecules because of the fast loss that occurs when two molecules approach at a close distance[8–10]. Here we demonstrate a new type of scattering resonance that is universal for a wide range of polar molecules. The so-called field-linked resonances[11–14] occur in the scattering of microwave-dressed molecules because of stable macroscopic tetramer states in the intermolecular potential. We identify two resonances between ultracold ground-state sodium–potassium molecules and use the microwave frequencies and polarizations to tune the inelastic collision rate by three orders of magnitude, from the unitary limit to well below the universal regime. The field-linked resonance provides a tuning knob to independently control the elastic contact interaction and the dipole–dipole interaction, which we observe as a modification in the thermalization rate. Our result provides a general strategy for resonant scattering between ultracold polar molecules, which paves the way for realizing dipolar superfluids[15] and molecular supersolids[16], as well as assembling ultracold polyatomic molecules.

Ultracold polar molecules with tunable dipole moments provide a powerful platform for quantum simulations[17,18], quantum computation[19,20] and ultracold chemistry[21]. Scattering resonances are long-sought-after tools in these systems, which have been essential in ultracold-atom experiments to control the contact interaction and for creating strongly correlated quantum phases[22], as well as for producing ultracold diatomic molecules[1]. Independent control over contact and long-range interactions in ultracold molecules has been predicted to enable the realization of new quantum phenomena such as exotic self-bound droplets and supersolid quantum phases[16]. Moreover, measurements of scattering resonances provide an accurate benchmark for calculations of the molecular potential energy surface[3,21] and open a new route in controlled quantum chemistry[5].

A scattering resonance occurs when the scattering state strongly couples to a quasibound state. Based on whether the quasibound state is hosted by the same or a different channel than the scattering channel, the resonance is categorized as a shape resonance or a Feshbach resonance, respectively. Shape and Feshbach resonances have been observed in atom–molecule and molecule–molecule collisions by scanning the collision energy by using molecular beams at kelvin and subkelvin temperatures[21,23–26]. In the ultracold (submicrokelvin) regime, scattering resonances are often induced by an external electromagnetic field that shifts the relative energy between the quasibound state and the scattering state[1]. Magnetically tunable Feshbach resonances have been observed in collisions between weakly bound Feshbach molecules[2,4] and recently between NaLi molecules in the spin-triplet ground state[6]. However, the magnetic tuning scheme essential to Feshbach resonances requires a non-zero electronic spin, and is, therefore, unlikely to find application for bialkali molecules in the spin-singlet ground state. The spin-singlet absolute ground state of bialkaline molecules is of special interest, as it is the only long-lived state in which the molecules feature strong electric dipole–dipole interactions (DDI). Moreover, Feshbach resonances are not expected to occur between ground-state molecules in the presence of nearly universal loss, owing to the high density of tetramer states near the collisional threshold and the loss mechanisms associated with collisional complexes[8–10]. A general method to realize collisional resonances of ultracold dipolar molecules, therefore, remains open.

Here we demonstrate a general approach to create such resonances in collisions between dipolar molecules by coupling them to so-called field-linked (FL) states[11,12]. These weakly bound states are induced by engineering an attractive well in the long-range intermolecular potential through microwave dressing[14,27]. Unlike in conventional resonances, in which an external field merely tunes an existing short-range quasibound state into resonance, the long-range FL states exist only in the presence of the microwave field. The sensitivity of the FL states to the microwave field leads to an unprecedented level of control over the intermolecular interaction. Here we demonstrate this tunability by observing two resonance branches in the inelastic scattering rate, the peak positions of which continuously shift with the microwave frequency and polarization. We further characterize the change of the thermalization rate caused by the diverging scattering length in a resonant collision channel.

[1]Max-Planck-Institut für Quantenoptik, Garching, Germany. [2]Munich Center for Quantum Science and Technology, München, Germany. [3]Institute for Molecules and Materials, Radboud University, Nijmegen, The Netherlands. [4]Fakultät für Physik, Ludwig-Maximilians-Universität, München, Germany. [5]These authors contributed equally: Xing-Yan Chen, Andreas Schindewolf. ✉e-mail: xinyu.luo@mpq.mpg.de

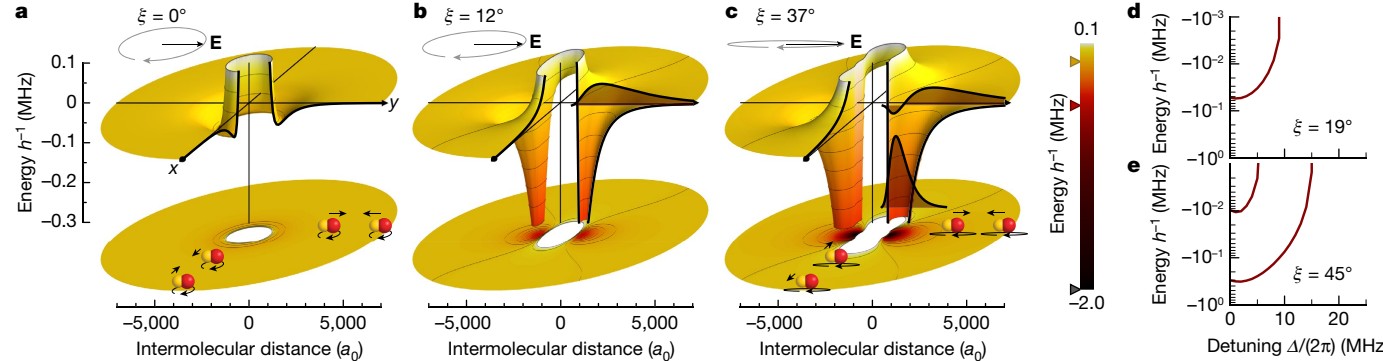

**Fig. 1 | Interaction potentials and bound states of microwave-dressed ground-state molecules. a–c**, The cut-open three-dimensional surfaces illustrate the interaction potentials $U(\mathbf{r})$ including the $p$-wave centrifugal potential between two molecules in the $x$–$y$ plane for different ellipticity angles, $\xi$, of the field polarization: $\xi = 0°$ (**a**), $\xi = 12°$ (**b**) and $\xi = 37°$ (**c**). Below, a projection of the same potential is shown. The interaction potential along the $z$ direction is always repulsive (not shown). The microwave is on resonance ($\Delta = 0$). The shaded areas in **b** and **c** show the radial wavefunction of the bound states. The insets visualize the rotating electric field vector **E**, and sketch the interaction between the rotating dipoles colliding along the $x$ or $y$ direction. The markers on the colour bar denote the potential depths for the three cases. **d,e**, Coupled-channel calculations of the energy of the bound states as a function of $\Delta$ for values of $\xi = 19°$ (**d**) and $\xi = 45°$ (**e**). In all panels the Rabi frequency is set to $\Omega = 2\pi \times 10$ MHz.

## Interaction potential

Polar molecules possess a permanent dipole moment $d_0$ in their body-fixed frame. To induce a dipole moment in the laboratory frame, external fields need to be applied to mix different rotational states and to break the rotational symmetry. Here we use microwave dressing between the two lowest rotational states of the molecules to polarize them. The induced dipole moment follows the alternating current electric field as $\mathbf{d}(t) = \overline{d}[\mathbf{e}_+(t)\cos\xi + \mathbf{e}_-(t)\sin\xi]$, where $\xi$ describes the ellipticity of the microwave radiation and $\mathbf{e}_\pm(t)$ are the $\sigma^\pm$ polarization basis vectors. The time-averaged dipole moment $\overline{d} = d_0/\sqrt{6(1 + (\Delta/\Omega)^2)}$ is tunable via the microwave detuning $\Delta$ and the Rabi frequency, $\Omega$. Because the rotational frequency of NaK at 5.643 GHz is much faster than all other dynamical timescales in the system, we consider the time-averaged DDI at long range[15]

$$U_{dd}(\mathbf{r}) = \frac{\overline{d}^2}{8\pi\epsilon_0 r^3}(3\cos^2\theta - 1 + 3\sin2\xi\sin^2\theta\cos2\varphi) \quad (1)$$

where $\mathbf{r} = (r, \theta, \varphi)$ is the relative position between the molecules in polar coordinates defined by the microwave wavevector. The microwave propagates in the $z$ direction, and the microwave polarization ellipse has its major and minor axes in the $y$ and $x$ direction. Remarkably, the symmetry of the interaction can be manipulated by the microwave ellipticity, as illustrated in Fig. 1a–c. For $\xi = 0°$ (circular polarization) and $\xi = 45°$ (linear polarization), where $U_{dd}$ resembles the typical DDI up to a constant prefactor[28]. In between linear and circular polarization, the interaction breaks the rotational symmetry along all directions.

As the molecules approach each other, microwave dressing induces an anisotropic van der Waals interaction $U_{vdW} \propto 1/r^6$ (Methods)[15]. With blue-detuned microwave dressing, $U_{vdW}$ is repulsive in all directions, which protects the molecules from loss processes at short range and reduces the inelastic cross-sections[14,29–31]. Such a shielding potential arises owing to an avoided crossing between the attractive and the repulsive branch of the DDI. In a semi-classical picture, this avoided crossing can be understood as the reorientation of colliding dipoles through DDI[31,32]. A similar flipping of the dipoles occurs between polar molecules in a direct current electric field[33,34], between Rydberg atoms[35] and between ions and Rydberg atoms[36].

The full interaction potential between two dressed molecules is the sum of the DDI potential and the van der Waals potential $U(\mathbf{r}) = U_{dd}(\mathbf{r}) + U_{vdW}(\mathbf{r})$. We can shape the interaction potential and control the scattering process, as illustrated in Fig. 1. Along the $y$ axis,

$U(\mathbf{r})$ resembles a Mie potential[15,37,38] with a characteristic length of about $10^3 a_0$. A deviation from circular polarization breaks the azimuthal symmetry of the DDI and enhances the depth of the potential well along the $y$ axis, which becomes deep enough to support one or more bound states[13]. These bound states are the FL states, the properties of which strongly depend on the external fields. By tuning the binding energy of the FL state across the collisional threshold, for example, with microwave detuning as shown in Fig. 1d,e, FL resonances occur, which drastically alter the scattering properties between the molecules.

The low-energy scattering in $U(\mathbf{r})$ can be described by the associated partial-wave phase shifts, which are given by[39]

$$\delta_{lm_l}(k) \approx (ka_{lm_l})^{2l+1} + kc_{lm_l}a_{dd} \quad (2)$$

Here $l$ and $m_l$ are the angular momentum and its projection along the quantization axis, $\mathbf{k}$ is the relative wavevector, $a_{lm_l}$ and $a_{dd}$ are the characteristic lengths associated with the contact interaction and the DDI respectively, and $c_{lm_l}$ denote partial-wave dependent prefactors for the dipolar scattering phase shifts. Note that the contact interaction is suppressed by the centrifugal barrier for $l \neq 0$ as $\mathbf{k} \to 0$, whereas the phase shift from the long-range DDI scales linearly with $\mathbf{k}$ in all partial waves, and is proportional to the dipolar length $a_{dd} = \mu\overline{d}^2/4\pi\epsilon_0\hbar^2$, where $\mu = m/2$ is the reduced mass of the molecule. The FL resonances provide a tuning knob for $a_{lm_l}$ in the resonant channel. By reducing $U_{vdW}$, FL resonances can occur at any desired dipole moment up to $d_0/\sqrt{6}$, thus realizing independent control of the contact interaction and the DDI.

## Resonance map

We map out the resonances by measuring the inelastic rate coefficient $\beta_{in}$ of collisions between the dressed molecules. The optically trapped ground-state $^{23}$Na$^{40}$K molecules with nuclear spin projections $(m_{i,Na}, m_{i,K}) = (3/2, -4)$ are formed from an ultracold atomic mixture by means of magnetoassociation and subsequent stimulated Raman adiabatic passage (STIRAP) at a magnetic offset field of 72.35 G (ref. [40]). For most measurements, the temperature $T$ of the molecular ensemble is 230 nK and the initial average density $n_0$ is about $5 \times 10^{11}$ cm$^{-3}$. Next, the microwave is ramped on in 100 µs to dress the molecules. After a variable hold time, the remaining molecules are released from the optical trap and we determine the number of molecules and the temperature from the time-of-flight images.

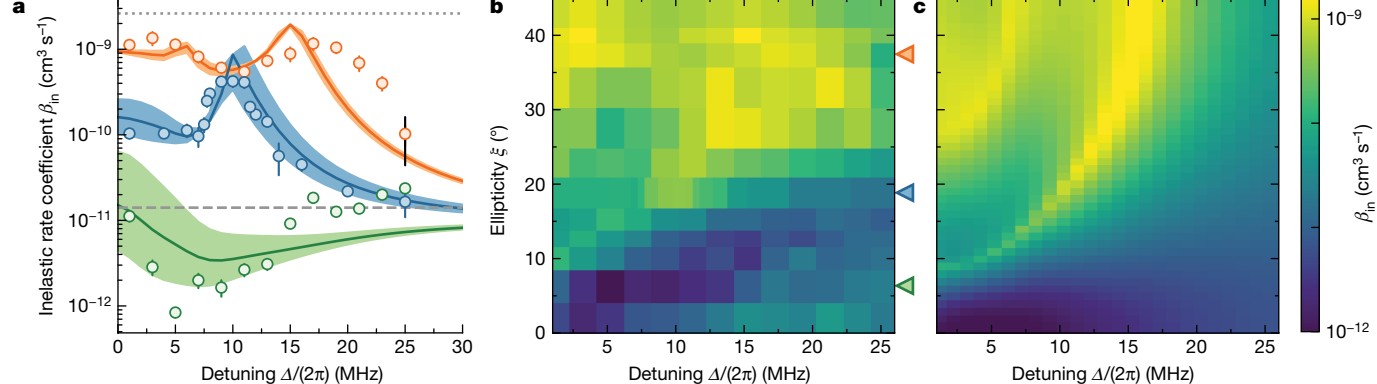

**Fig. 2 | FL resonances. a**, Inelastic collision rate coefficient $\beta_{in}$ between microwave-dressed NaK molecules as a function of the microwave detuning $\Delta$ for various microwave polarizations with the ellipticity angle $\xi = 6(2)°$ (green), $19(2)°$ (blue) and $37(2)°$ (orange) at the Rabi frequency $\Omega \approx 2\pi \times 10$ MHz. The solid lines show the corresponding theory calculations. The shaded regions show the calculations within the uncertainty of $\xi$. The grey dashed line denotes the theoretical universal value of $\beta_{in}$ and the grey dotted line denotes the single-channel unitarity limit. The coloured error bars show the standard deviation of the fit results and the black error bar illustrates the exemplary common systematic uncertainty. **b,c**, Colour density map of the experiment data (**b**) and the theory calculation (**c**) of the inelastic rate coefficient as a function of microwave detuning and ellipticity. The triangles on the right axis of **b** mark ellipticity for the data shown in **a**.

As we tune the ellipticity of the microwave from circular to linear, up to two FL states emerge from the dressed potential. Figure 2a shows exemplary loss rate coefficients for three different ellipticities. At $\xi = 6(2)°$, the potential is too shallow to support a bound state, thus no resonance is observed. In this regime, the inelastic collision is suppressed by the shielding potential at small detunings[31]. For $\xi = 19(2)°$, the interaction potential supports a single bound state near zero microwave detuning, leading to enhanced inelastic scattering at $\Delta \approx 2\pi \times 10$ MHz. For $\xi = 37(2)°$, the potential becomes deep enough to support two bound states, leading to two resonance peaks.

One special feature of the FL resonances is their sensitivity to external fields. We show that the resonance position continuously changes with the microwave parameters by mapping out the two resonance branches while varying the microwave detuning and polarization. Figure 2b shows two branches of FL resonances, starting at $\xi \approx 10°$ and $\xi \approx 32°$. As the polarization ellipticity $\xi$ increases, less DDI is needed to support the bound states and the resonances, therefore, shift to larger detuning. However, the global inelastic rate coefficient increases as the polarization becomes more elliptical, because of the increased coupling to the other dressed states[28]. Overall our measurements show good agreement with our theory predictions (Fig. 2c). We attribute the broadening and shift of the resonance peaks compared to the theory to an increase of the Rabi frequency as we scan the detuning (Methods). These systematic errors affect mostly the FL resonances near linear polarization, where the potential depth is more sensitive to the relative detuning.

## Temperature dependence of the inelastic scattering

The temperature dependence of the inelastic scattering rate varies with the detuning. At large detuning where the DDI is reduced, the inelastic scattering rate is universal[41] and scales for identical fermions linearly with temperature. At small detuning, the scattering enters the semi-classical regime for which $\beta_{in}$ is independent of the temperature[42]. On the scattering resonance, the collision rate has a temperature dependence that is reminiscent of the unitarity limit[5,32], whereas the loss remains substantially smaller than this limit because of shielding. Meanwhile, the width of the resonance feature is broadened by thermal averaging. As a consequence, for temperatures as high as 700 nK, the resonance becomes less visible as shown in Fig. 3. When the collision energy becomes lower than the centrifugal barrier of the interaction potential, the resonance peak would be further narrowed owing to

the increased lifetime of the quasibound state. Therefore, reaching ultracold temperatures is crucial for the observation of FL resonances.

## Elastic scattering

Scattering resonances are associated not only with enhanced losses of the molecules, but more importantly, the ability to control elastic scattering. With FL resonances, we can tune the elastic scattering rate while keeping the inelastic rate small.

We characterize the effect of the FL resonances on the elastic collision rate by measuring the thermalization rate of the samples. This is commonly done by quenching the trapping confinement in one dimension and observing the global cross-dimensional thermalization[43]. However, for small $\Delta/\Omega$ our samples are in the hydrodynamic regime, in which the global thermalization rate is limited by the trapping frequencies[31]. Instead, we perturb the momentum distribution of the molecular cloud by pulsing on an optical-lattice beam for 300 μs. The lattice pulse diffracts some molecules and sends them to collide along the y axis, defined as the long-axis of the microwave field (Fig. 1), along which the DDI is most attractive. Fast local thermalization smears out the diffraction pattern that is formed in momentum space during

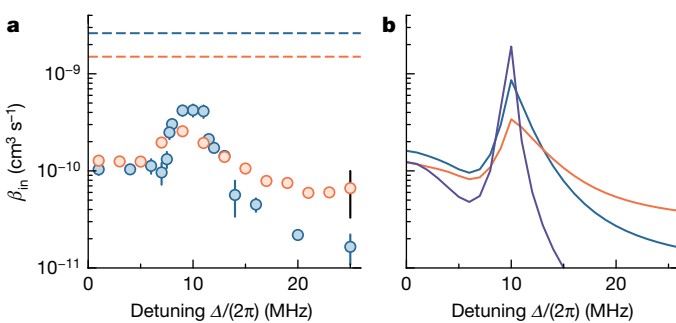

**Fig. 3 | Temperature dependence of the inelastic scattering. a,b**, Experimental results (**a**) and calculations (**b**) of the inelastic collision rate coefficient $\beta_{in}$ as a function of the microwave detuning $\Delta$ at 700 nK (orange), 230 nK (blue) and 20 nK (purple). The coloured error bars show fitting errors, and the black error bar additionally contains the common systematic uncertainty. The solid lines are coupled-channel calculations and the dashed lines are the unitarity limit at the corresponding temperatures. The molecules are dressed by microwaves with ellipticity $\xi = 19(2)°$ and Rabi frequency $\Omega \approx 2\pi \times 10$ MHz.

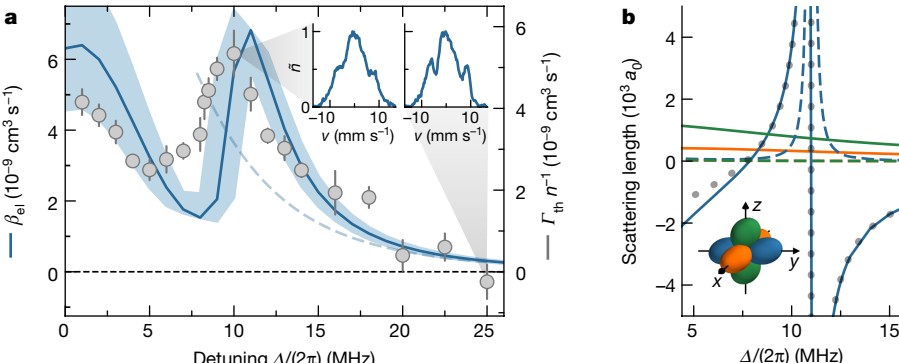

**Fig. 4 | Elastic scattering. a**, The grey data points show the thermalization rate $\Gamma_{th}$ normalized by the mean in situ density $n$ as a function of the microwave detuning $\Delta$ at an ellipticity angle $\xi = 19(2)°$ and a Rabi frequency $\Omega \approx 2\pi \times 10$ MHz. The temperature is 230 nK. The error bars are the standard error of the mean of 7 to 16 repetitions. For comparison, the solid blue line shows the corresponding theory calculation of the elastic collision rate coefficient $\beta_{el}$. The uncertainty of $\xi$ is taken into account by the shaded area. The dashed line is the Born approximation of the background collision rate coefficient, which holds for detunings $\Delta \gtrsim 2\pi \times 8$ MHz (Methods). The insets show the normalized linear density $\tilde{n}$ along the lattice axis averaged over ten repetitions as a function of the molecule velocity $v$ in the lattice direction for $\Delta = 2\pi \times 10$ MHz and $2\pi \times 25$ MHz. **b**, The coupled-channel calculations of the energy-dependent scattering length with the same microwave parameters and a fixed collision energy of $k_B T$, with $T = 230$ nK. The solid (dashed) lines are the real (imaginary) part of the scattering lengths in the channel $p_x$ (orange), $p_y$ (blue) and $p_z$ (green). The dotted line is a fit of equation (3). The inset illustrates the partial waves of the scattering channels. $k_B$ is the Boltzmann constant.

the lattice pulse (insets in Fig. 4a). From the contrast of the diffraction pattern we can estimate the thermalization rate $\Gamma_{th}$ (Methods).

We observe that the measured thermalization rate follows a similar trend as the calculated value of the elastic collision rate coefficient, $\beta_{el}$ (Fig. 4a). Besides the contribution from the DDI, which scales with $\bar{d}^2$ and decreases as $\Delta$ increases, a clear resonance feature is visible around the FL resonance. The shift of the resonance between the experiment data and theory is within the systematic uncertainty of the ellipticity. The average number of elastic collisions that is needed per particle to reach thermalization is so far unknown in the regime of elliptical microwave polarization. From the comparison between the measured $\Gamma_{th}/n$ and $\beta_{el}$ from our coupled-channel calculations, we find that this factor is close to 1 under the present experimental conditions.

The observed elastic scattering rate has contributions from both contact interaction and long-range DDI. As shown in equation (2), the DDI contributes in multiple channels, whereas the contact interaction only has a notable contribution in the resonant channel, owing to its unfavourable scaling with the wavevector. Although such an interplay between contact interaction and DDI limits the change in the total elastic scattering rate, the underlying scattering length, however, shows divergent behaviour in the resonant channel. Figure 4b shows the energy-dependent scattering length $\tilde{a}_{lm_l}(k) = -\tan\delta_{lm_l}(k)/k$ (refs. [41,44]) for the three $p$-wave channels at the average collision energy. The real (imaginary) part of the scattering length corresponds to the elastic (inelastic) scattering. The FL resonance occurs in the $p_y$ channel where the interaction is most attractive. The corresponding scattering length shows a resonance feature, in which the real part can be tuned to large positive or negative values, whereas the imaginary part remains small. The ratio of elastic-to-inelastic collisions is about ten on the resonance, and can be further enhanced at smaller ellipticity, in which the resonance shifts towards higher Rabi frequencies (Methods).

A simple analytic formula that describes the resonant elastic scattering length is given by

$$\tilde{a}_{1y} = a_{dd}\left(-\frac{1 + 3\sin2\xi}{10}\right)\left(1 + \frac{\Delta^*(k)}{\Delta - \Delta_0(k)}\right) \quad (3)$$

where $\Delta_0(k)$ and $\Delta^*(k)$ denote the position and the width of the resonance. The width $\Delta^*(k) \propto k^2$ follows the scaling of the $p$-wave contact interaction. For the collision energy considered here, we extract $\Delta_0 \approx 2\pi \times 10.99$ MHz and $\Delta^* \approx 2\pi \times 3.29$ MHz from the fit to the coupled-channel calculations.

The resonance position $\Delta_0(\mathbf{k})$ also has a weak energy dependence. As a consequence, the resonance position with thermal averaging is broadened and slightly shifted towards lower detuning. These thermal effects, however, will be greatly suppressed in a degenerate Fermi gas, in which the scattering predominantly occurs near the sharp Fermi energy[15].

## Discussion

Field-linked resonances provide a new universal tool to control the collisions between ultracold polar molecules. These resonances occur as long as the Rabi frequency is sufficiently large, such that the interaction potential is deep enough to support the bound states. Quantitative predictions of FL resonances only require knowledge of the mass, the dipole moment and the rotational structure of the individual molecules as well as their loss rate at short range. This is in stark contrast to molecular collisions involving close contact between the molecules, in which a large number of collision channels are involved and the existing knowledge of the potential energy surface are too imprecise to predict the number of bound states, let alone their position. Precise knowledge on the FL states also makes them useful as intermediate states in photoassociation spectroscopy to probe the short-range potential[11].

The control over the scattering length opens up new possibilities to investigate many-body physics with both contact interaction and DDI. In a degenerate Fermi gas, the resonant interaction facilitates realization of dipolar superfluidity[27,45,46]. Specifically, pairing between molecules is enhanced because of the presence of the FL bound state. Therefore, the critical temperature for Bardeen–Cooper–Schrieffer superfluidity increases drastically near the FL resonance, that is, to about 14% of the Fermi temperature for NaK molecules[15]. The anisotropic nature of such a dipolar superfluid gives rise to new quantum phenomena such as gapless superfluidity[47] and topological $p_x + ip_y$ symmetry[48]. In a Bose–Einstein condensate (BEC), independent control over the $s$-wave scattering length and dipolar length has led to the observation of self-bound droplets and the formation of supersolids in magnetic atoms[49]. Making use of FL resonance with bosonic polar molecules, the dipolar lengths of which are orders of magnitude larger than magnetic atoms, would enable the study of such exotic phenomena in entirely unexplored regimes[16].

The observed resonances also demonstrate the existence of the FL states, a new set of exotic long-range polyatomic molecular states.

Tetramer molecules with approximately twice the dipole moment of the individual diatomic molecules could potentially be created by adiabatically ramping the microwave field across a FL resonance or by radio-frequency association. Those composite bosonic tetramers are expected to be long lived at small binding energies[12] and could be collisionally stable owing to the shielding between the constituent molecules. Below the critical temperature, a tetramer gas could form a BEC[50] and may lead to a new crossover from a dipolar Bardeen–Cooper–Schrieffer superfluid to a BEC of tetramers.

## Conclusion

We have observed a new type of universal scattering resonance between ultracold microwave-dressed polar molecules that is associated with FL tetramer bound states in the long-range potential well. The resonances are highly tunable via microwave power, frequency and polarization, which makes them a versatile tool for controlling molecular interactions. Because the FL states are insensitive to species-dependent short-range interactions, the FL resonance is applicable to a wide range of polar molecules. Our results provide a general route to strongly interacting molecular gases and open up new possibilities to investigate new quantum many-body phenomena and to produce long-lived dipolar tetramer molecules.

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

## Methods

### Microwave field generation

We use a dual-feed square-waveguide antenna to generate a microwave field with tunable polarization. The waveguide is fabricated from copper-coated glass-fibre-reinforced epoxy laminate. The inner dimensions of the waveguide are $33 \times 33 \times 58$ mm. The width of the waveguide is chosen such that the cut-off frequency of the transverse electric ($\text{TE}_{10}$) mode is below the rotational transition frequency of the molecules at 5.64 GHz and that the microwave field strength at the position of the molecules is optimal. To achieve impedance matching, the two feeds are 13 mm long, which is a quarter wavelength in free space. They are placed orthogonal to each other, and 22 mm away from the back plate of the waveguide, which is about a quarter wavelength in the waveguide. Each feed produces a close to linearly polarized electric field at the position of the molecules, which is about 25 mm away from the top of the waveguide. When the field strengths of these subfields are balanced, tuning the relative phase of the feeds allows straightforward tuning of the field polarization. A relative phase of approximately 90° (−90°) results in a $\sigma^+$ ($\sigma^-$) polarized field, whereas approximately 0° (180°) produces a linearly polarized field along the $y$ ($x$) direction.

The microwave setup is sketched in Extended Data Fig. 1. A Rohde & Schwarz SMA100B signal generator (using a yttrium–iron–garnet oscillator) with the noise suppression option SMAB-B711 is used as microwave source. To independently control the two feeds of the antenna, the microwave is divided by a power splitter into two paths. Each path includes a voltage-controlled attenuator to balance the subfields, a 10-W amplifier (KUHNE electronic KU PA 510590-10 A) and a mechanical phase shifter (SHX BPS-S-6-120) for differential phase control.

### Calibration of the field polarization

To characterize the polarization of the microwave field, we initially measure its $\pi$, $\sigma^+$ and $\sigma^-$ component in the frame of the magnetic offset field. We do this at low microwave power by measuring the Rabi coupling to excited rotational states with different projections $m_J$ of the rotational quantum number $J$ on the magnetic field axis [31]. A direct measurement of the $\sigma^+$ and $\sigma^-$ component at large microwave power is impracticable, as the Rabi frequency is then orders of magnitude larger than the Zeeman splitting of the $m_J$ states. When we change the relative phase $\phi$ between the feeds, each field component, and thus each Rabi frequency, individually oscillates with a period of 360° owing to the interference between the imperfect subfields, as shown in Extended Data Fig. 2a. We fit these oscillations with the function

$$\Omega(\phi) = \sqrt{\Omega_1^2 + \Omega_2^2 + 2\Omega_1\Omega_2\cos(\phi + \phi_0)} \qquad (4)$$

with the fit parameters $\Omega_1$, $\Omega_2$ and $\phi_0$, which define the contributions of the individual feed. The offset phases $\phi_0$ have an uncertainty of 2.9°, which we attribute to the hysteresis and imperfect tuning of the mechanical phase shifters. Note, for the calibration measurements presented in Extended Data Fig. 2a, the power balance between the feeds was tuned to minimize the $\sigma^-$ component around $\phi = 90°$. The finite ellipticity of the subfields causes the field strengths at other angles of $\phi$ to be unbalanced, so that we do not get pure $\sigma^+$ polarization at $\phi = -90°$ and the phase values that provide linear polarization shift away from 0° and 180°.

We calculate the ellipticity $\xi$ from the fitted Rabi frequencies for each individual relative phase, as shown in Extended Data Fig. 2b. At high microwave power, the electric field of the microwave replaces the magnetic offset field as the quantization axis, so that $\xi$ is defined in the frame of the microwave field. Because the relative phase between the measured field components is unknown, we can only deduce the limits for the tilt angle of the microwave wavevector with respect to the magnetic offset field, which causes a systematic uncertainty of $\xi$. Close to $\sigma^+$ polarization the tilt angle is only around 10°, so that the uncertainty of $\xi$ is dominated by the above-mentioned uncertainties of $\phi_0$.

When switching from the calibration measurements at low microwave power to the measurements at high power, we have to consider the non-linearity of the amplifiers close to their saturation power. This shifts the power balance between the two antenna feeds when we increase the microwave power and thereby changes $\xi$. We reestablish the power balance by tuning one voltage-controlled attenuator while optimizing the shielding at $\phi = 90°$, where we initially minimized $\xi$. We find that we need to compensate the relative power by 10%. In addition, we also observe a small variation of the relative phase and power as we scan the microwave detuning. The variation in the relative phase leads to a systematic uncertainty of the ellipticity on the same order as the contribution from the uncertainty of $\phi_0$. The variation in Rabi frequency could broaden the resonance feature, especially for ellipticities close to linear polarization, as observed in Fig. 2.

### Inelastic collision rate coefficient

The inelastic collision rate coefficients $\beta_{in}$ are experimentally determined from the time evolution of the measured molecule number $N$ and the temperature $T$ by numerically solving the differential equations[51]

$$\frac{dN}{dt} = (-\beta_{in}n - \Gamma_1)N \qquad (5)$$

$$\frac{dT}{dt} = h \qquad (6)$$

with the average density

$$n = \frac{N}{8\sqrt{\pi^3 k_B^3 T^3/m^3 \overline{\omega}^6}} \qquad (7)$$

where $\overline{\omega}$ is the geometric mean trapping frequency and $\Gamma_1$ is the one-body loss rate. We assume both the heating rate and the two-body loss rate coefficient to be constant during the loss process. As the overall heating is no more than 50%, the fitted values of $\beta_{in}$ do not significantly change when we instead assume a linear temperature dependence of the rate coefficient.

The comparison between loss measurements on the scattering resonance and away from resonance are shown in Extended Data Fig. 3a. To limit the number of free fit parameters, we determine $\Gamma_1 = 0.53(2)$ Hz in independent measurements at low densities, as shown in Extended Data Fig. 3b.

### Fast thermalization measurements

The momentum distribution of the molecular cloud is disturbed by pulsing on a one-dimensional optical lattice for $t_{lat} = 300$ μs. Subsequently the microwave power is ramped down in 100 μs and the trapping confinement is turned off. After 10 ms time of flight, the momentum distribution is imaged. The lattice is approximately aligned along the $y$ direction. Its lattice constant is $a_{lat} = 532$ nm and the lattice depth is $88E_r$, where $E_r = h^2/(8ma_{lat}^2)$ is the lattice recoil energy. The pulse duration was chosen to optimize the contrast of the resulting interference pattern. Note that the pulse is short compared to the trap frequencies $2\pi \times (82, 58, 188)$ Hz of the background confinement, so that cross-talk between the momentum distribution and the real-space density is small. Also two-body loss, even on resonance, is negligible on this time scale.

To estimate the degree of thermalization $c_{th}$ we make the simplifying assumption that the momentum distribution after the lattice pulse can be described by

$$\tilde{n}(v) = c_{\mathrm{th}}\tilde{n}_{\mathrm{th}}(v) + (1 - c_{\mathrm{th}})\tilde{n}_0(v) \tag{8}$$

where $\tilde{n}_{\mathrm{th}}(v)$ is a Gaussian (that is, thermal) distribution and $\tilde{n}_0(v)$ is the undisturbed interference pattern (Extended Data Fig. 4a). We determine $\tilde{n}_0(v)$ by averaging over 16 images of the interference pattern in the absence of microwave-induced interactions. We further assume that the interference pattern decays exponentially, such that the thermalization rate is given by

$$\Gamma_{\mathrm{th}} = -\ln(1 - c_{\mathrm{th}})/t_{\mathrm{lat}} \tag{9}$$

Given the average number of elastic collisions that is required to reach thermal equilibrium $N_{\mathrm{col}}$ and the average relative velocity $v_{\mathrm{rel}}$ of the colliding molecules, the elastic scattering cross-section is given by

$$\sigma_{\mathrm{el}} = \frac{N_{\mathrm{col}}\Gamma_{\mathrm{th}}}{nv_{\mathrm{rel}}} \tag{10}$$

Assuming that the thermalization is mainly driven by elastic collisions between molecules in the side peaks and molecules in the main peak of the diffraction pattern, the average relative velocity can be approximated as

$$v_{\mathrm{rel}} = \sqrt{\bar{v}^2 + h^2/(ma_{\mathrm{lat}})^2} \tag{11}$$

where $\bar{v} = \sqrt{16k_{\mathrm{B}}T/(\pi m)}$ is the thermally averaged collision velocity in the undisturbed sample. The expression of $N_{\mathrm{col}}$ is so far unknown in the regime of elliptical microwave polarization. We can, however, set an upper limit by assuming $\xi = 0$. In that case, $N_{\mathrm{col}} \approx 4$ for a tilt angle of the microwave field of 10° (refs. [34,43]). A comparison of this simplified model with our predictions of $\beta_{\mathrm{el}} = \bar{v}\sigma_{\mathrm{el}}$ is shown in Extended Data Fig. 4b.

### Born approximation
The elastic scattering rate (Fig. 4a) from the DDI is determined by the long-range $U_{\mathrm{dd}}$. In the low-energy regime $E \lesssim \hbar^2/\mu a_{\mathrm{dd}}^2$, the elastic scattering rate can be obtained via the Born approximation $\beta_{\mathrm{el}} = \sigma_{\mathrm{el,Born}}\bar{v}$, where

$$\sigma_{\mathrm{el,Born}} = \frac{16\pi}{15}(1 + 3\sin^2 2\xi)a_{\mathrm{dd}}^2 \tag{12}$$

is the elastic cross-section[42].

We can also obtain the dipolar scattering lengths from the Born approximation. Because $U_{\mathrm{dd}}$ is symmetric under reflection along the three Cartesian axes, the $p_x$, $p_y$ and $p_z$ channels are decoupled and the main contributions are the elastic scattering within each channel. The corresponding scattering lengths are given by

$$\tilde{a}_{1x,\mathrm{Born}} = -\frac{1}{10}(1 - 3\sin 2\xi)a_{\mathrm{dd}} \tag{13}$$

$$\tilde{a}_{1y,\mathrm{Born}} = -\frac{1}{10}(1 + 3\sin 2\xi)a_{\mathrm{dd}} \tag{14}$$

$$\tilde{a}_{1z,\mathrm{Born}} = \frac{1}{5}a_{\mathrm{dd}} \tag{15}$$

The scattering length given in equation (3) is $\tilde{a}_{1y,\mathrm{Born}}$ plus the contribution from the FL resonance.

### Resonance near circular microwave polarization
The elastic-to-inelastic collision ratio near the FL resonances can be improved with better shielding near the circular microwave polarization and with lower temperature. The ratio is then about 900 at the

maximum elastic scattering rate and about 130 at the scattering resonance, as shown in Extended Data Fig. 5. Under these conditions, the FL resonance occurs at a much higher Rabi frequency compared to the observed resonances at more elliptical polarizations. However, this is still realistic to achieve by using an improved antenna design with increased microwave power.

### Induced van der Waals interaction
Reference[15] provides an analytical formula for the induced van der Waals interaction

$$U_{\mathrm{vdW}}(\mathbf{r}) = \frac{35C_{6\mathrm{i}}}{4r^6}\sin^2\theta(\cos^2\theta + 1$$
$$- 2\sin 2\xi \cos^2\theta \cos 2\varphi - \sin^2 2\xi \sin^2\theta \cos^2 2\varphi) \tag{16}$$

where the induced $C_{6\mathrm{i}}$ coefficient is given by

$$C_{6\mathrm{i}} = \frac{d_0^4}{1120\pi^2 \epsilon_0^2 \Omega (1 + (\Delta/\Omega)^2)^{3/2}} \tag{17}$$

For $\Omega = 2\pi \times 10$ MHz and zero detuning, $C_{6\mathrm{i}} = 1.1 \times 10^{-72}$ kg m$^8$ s$^{-1}$ is orders of magnitude larger than the direct isotropic van der Waals coefficient $C_6 = -4.9 \times 10^{-74}$ kg m$^8$ s$^{-1}$ (ref. [52]). The induced $C_{6\mathrm{i}}$ reduces as the detuning increases and can be comparable to $C_6$ for $\Delta/\Omega > 2.7$.

### Coupled-channel calculations
We perform coupled-channel scattering calculations by using the framework described in detail in refs. [28,29]. Here we summarize the numerical details of these calculations.

The NaK molecules are described as rigid rotors with the rotational states $J = 0, 1$. Interactions of the molecules with elliptically polarized microwaves are included as described in ref. [53]. The wavefunction for the relative motion is expanded in partial waves $L = 1, 3, 5$. Hyperfine interactions are not included as these were previously found to have a negligible effect[31] when operating at sufficiently large magnetic fields[29,53]. The colliding molecules interact with one another through the DDI, and undergo short-range loss, which is modelled as a capture boundary condition imposed at $r = 20a_0$. We propagate the coupled-channel equations outwards to $r = 10^6 a_0$ and match the solution to the scattering boundary conditions. This yields the scattering matrix, from which collision cross-sections and rate coefficients are determined and thermally averaged by using an energy grid of 21 energies that are logarithmically spaced between $0.03k_{\mathrm{B}}T$ and $32k_{\mathrm{B}}T$.

Binding energies of the FL bound states are calculated as follows. First, we compute adiabatic potentials by diagonalizing the Hamiltonian described above excluding the radial kinetic energy for fixed values of the molecule–molecule separation $r$. On each adiabatic potential curve, we compute bound states by using a sinc-function discrete variable representation[54]. We find that the position of the zero-energy bound states computed in this approximation agree well with the resonance positions, indicating that one can think of the FL bound states as living on a single adiabatic potential curve. Both resonances found here are supported by the lowest adiabatic potential, that is, the second resonance corresponds to a radial vibrational excitation, rather than an angular excitation. Interaction potentials that are shown in Fig. 1 are similarly computed as adiabatic potential curves, except that these are computed for a fixed orientation of the intermolecular axis relative to the microwave polarization, rather than by using a partial wave expansion.

Note, the coupled-channel calculations have no free parameters.

### Data availability
The experimental data that support the findings of this study are available from the corresponding author upon reasonable request. Source data are provided with this paper.

## Code availability

All relevant codes are available from the corresponding author upon reasonable request.

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

**Acknowledgements** We thank T. Shi for stimulating discussions and providing the analytical formula for the interaction potential, J. Hutson, R. R. W. Wang, Y. Bao and H. Adel for stimulating discussions, and C. Buchberg and M. Hani for the cooperation on the development of the waveguide antenna. We gratefully acknowledge support from the Max Planck Society, the European Union (PASQuanS grant no. 817482) and the Deutsche Forschungsgemein–schaft under Germany's Excellence Strategy–EXC-2111–390814868 and under grant no. FOR 2247 and MCQST seed funding program. A.S. and T.H. acknowledge funding from the Max Planck Harvard Research Center for Quantum Optics.

**Author contributions** All authors contributed substantially to the work presented in this manuscript. X.-Y.C., A.S. and S.E. carried out the experiments and together with R.B., M.D. and S.B. improved the experimental setup. X.-Y.C., A.S. and S.E. analysed the data. T.K. performed the theoretical calculations. T.H., I.B. and X.-Y.L. supervised the study. All authors worked on the interpretation of the data and contributed to the final manuscript.

**Funding** Open access funding provided by Max Planck Society.

**Competing interests** The authors declare no competing interests.

**Additional information**
**Correspondence and requests for materials** should be addressed to Xin-Yu Luo.

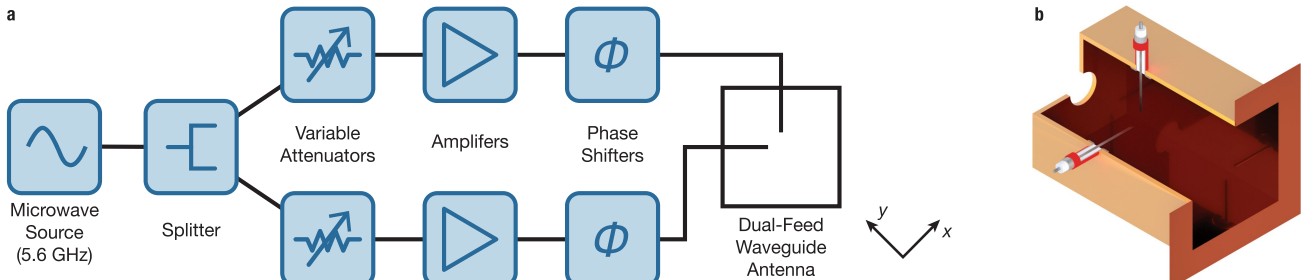

**Extended Data Fig. 1 | Microwave setup. a**, Electronic setup used to generate and control the microwave field. Individually, the two antenna feeds produce mainly linear polarized fields parallel to each feed, respectively. The voltage-controlled attenuators are used to balance the fields and to adiabatically ramp the field intensity. The phase shifters allow to tune the polarization. **b**, Half-section view of the waveguide antenna that shows the inside of the waveguide and the transition from the coaxial cable (red jacket) to the feed.

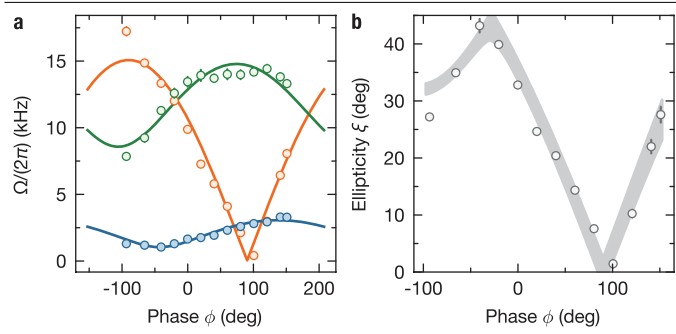

**Extended Data Fig. 2 | Calibration of the field polarization. a**, The Rabi frequencies of rotational $\sigma^+$ (green), $\pi$ (blue), and $\sigma^-$ (orange) transitions at low microwave power as a function of the phase shift $\phi$ between the antenna feeds. The error bars show the fitting error of the Rabi oscillations. The solid lines are fits to equation (4). **b**, Ellipticity of the microwave field in the frame of the microwave. The data points show the ellipticity angle $\xi$ calulated from the data in **a**. The error bars denote the uncertainty of $\xi$ that originates from the projection from the frame of the magnetic offset field to the frame of the microwave field due to the unknown phase relation between the three field components. The gray band is calculated from the fit functions shown in **a** and its width considers the uncertainty of the microwave orientation and the uncertainty of the offset phases $\phi_0$.

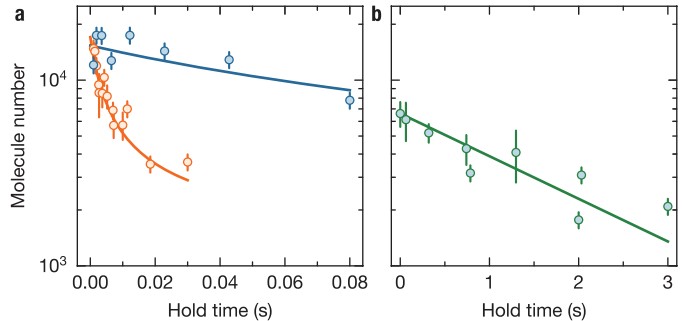

**Extended Data Fig. 3 | One- and two-body loss. a**, Example molecule loss at $\xi \approx 19°$ with detuning on resonance $\Delta = 2\pi \times 10$ MHz (bright) and away from resonance $\Delta = 2\pi \times 25$ MHz (dark). The lines are fits to the differential-equation model. **b**, Loss at low initial densities from which the one-body loss rate $\Gamma_1$ is determined. The line is an exponential fit function. The error bars show the standard deviation for repeated experiments.

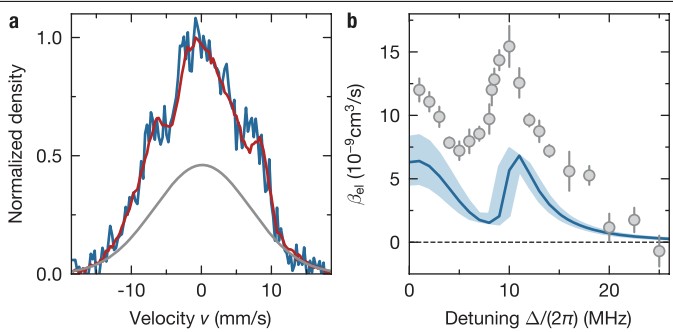

**Extended Data Fig. 4 | Thermalization model. a**, The blue line shows the diffraction pattern $\bar{n}(v)$ from a single experiment run at $\Delta = 2\pi \times 10$ MHz. The red line is a fit to equation (8). The gray curve describes the thermalized part $c_{th}\bar{n}_{th}(v)$. **b**, The same measurement as in Fig. 4a but here a simplified model is used to determine $\beta_{el}$ from the experimental data. The error bars show the standard error of the mean of 7–16 repetitions.

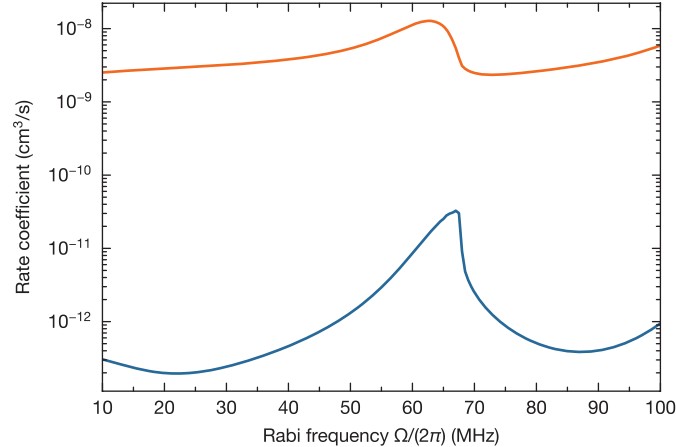

**Extended Data Fig. 5 | Field-linked resonance near circular microwave polarization.** Coupled-channel calculations of the elastic (orange) and inelastic (blue) scattering rate coefficient at $T = 20\,\text{nK}$, $\xi = 1°$, and $\Delta = 2\pi \times 1\,\text{MHz}$.