## [Peer Review File · Nature]

Manuscript Title: Field-linked resonances of polar molecules

Reviewer Comments & Author Rebuttals

Reviewer Reports on the Initial Version:

Referees' comments:

Referee #1 (Remarks to the Author):

This paper presents experiment and theory regarding the observation of a predicted new class of “field-linked” states of two weakly bound polar diatomic molecules induced by a resonant microwave field. The existence of such states due to a static electric field was predicted nearly 2 decades ago (Refs. 11,12) and more recently due to an oscillatory electromagnetic field {Ref. 14}. The field produces a long-range potential that accomplishes two desired outcomes: effective shielding against destructive short-range collisions and the binding of a field-tunable long-range bound tetramer state that makes large changes in the scattering properties of the two molecules. The present paper experimentally measures the properties of such states using the $J = 0$ to 1 rotational transition of the ground state of the NaK molecule induced by microwave radiation and carries out theoretical calculations that account for the measurements and aid in interpreting them. Measured and calculated properties are in semi-quantitative or at least qualitative agreement, indicating that the understanding of the experiment is correct. Since the experiment used identical fermionic molecules, the collisions were three-fold degenerate p-wave ones, with a resonance in one of the p-wave channels; resonances are also expected in s-wave channels of identical bosons or non-identical species. Such field-linked resonances, which are of a generic type to be expected for a range of molecules, give an additional experimental tool by which to manipulate the elastic and inelastic collisional properties of ultracold polar molecules, hopefully leading to new applications in few- and many-body physics and quantum information. There is also a hope, supported by theoretical calculations, that coherent elastic collisions will predominate over lossy inelastic ones by a sufficient amount to enable practical lossless microwave manipulation of molecular quantum gases or lattices.

The results are sound, represent a significant experimental advance demonstrating this new type of microwave-tunable resonance control of ultracold molecular collisions, and are worthy of publishing in Nature.

I found the paper clear in its presentations and graphics. I do not see a need for any significant revisions. I have just a couple points to note that the authors might wish to comment upon.

Looking through the paper again a second time, I could not find which particular hyperfine state of the molecule the authors used. Perhaps this was my oversight, but for the record, the state should be mentioned in the text, even if hyperfine effects turn out to be negligible. Given the great complexity of the hyperfine structure of the field-dressed tetramer potentials and the likelihood of avoided crossing of field-dressed states coupled by inherent potential anisotropies (see Ref. 12 for

the static field case), I wonder why hyperfine effects are negligible? The authors do mention that the (previously published) theory suggest that they are. I gather that the mostly isotropic long range shielding repulsion prevents any tunneling to shorter distances where lossy collisions might occur. The losses that are observed here and are predicted then would be due to very-long range couplings among the field-dressed states at distances larger than the shielding repulsion, although it is not clear why hyperfine does not play a role, given the small sub-MHz nuclear spin splitting in a singlet-Sigma molecular ground state (much smaller than the Rabi frequency or detunings used). Any additional enlightenment on this here would be helpful.

The authors use the asymptotic field-dressed van der Waals coefficient for their potentials. I presume this field-dressed repulsive contribution must be large in magnitude compared to the direct attractive van der Waals coefficient, which is known to have a large dipolar contribution for two NaK molecules. They might mention the relative sizes of the direct and field-dressed contributions to justify the neglect of the ordinary term under their conditions.

Referee #2 (Remarks to the Author):

This is an excellent manuscript reporting the first observation of Feshbach resonances in ultra-cold ground state molecular collisions induced by microwave field. Feshbach resonances have been observed recently in many different systems, however collisions between alkali molecules pose a special problem. At the short range they lead to reaction with nearly unit probability. As such, existence of scattering resonances should be strongly suppressed. In this beautiful experimental work authors build upon ideas of microwave shielding and in particular reference [14]. Microwave radiation couples between two rotational states and generates an adiabatic shallow potential well that supports several bound states. Authors with experimentally and theoretically prove that such states indeed exist, with their energy position depending on microwave field ellipticity and detuning. Both parameters can be used in order to tune the position of a quasi-bound state to match the collision energy. This is a very important achievement, it opens a new window into experiments with polar ultra-cold molecules with tuneable interactions despite the near perfect chemical reactivity that has hindered progress for many years.

Referee #3 (Remarks to the Author):

The manuscript by X.-Y. Chen et al. reports an experimental investigation of a collisional resonance between ultracold ground-state NaK molecules. These so-called field-linked resonances arise for microwave-dressed molecules when the engineered intermolecular attractive potential well is sufficiently deep and supports (in this case) one or two bound states.

This work pushes ultracold molecular physics to its experimental limits, and reveals a new and quite convincing method to alter molecular interactions on a significant scale. It comes at a time of a rapid

progress in quantum control of molecules and is sure to solicit wide-ranging interest from readers. In addition, this method could be applicable more generally and is not expected to be specific to the NaK system. The authors reasonably claim that applications of this control method are not far off.

The paper is well written, using a clear and accessible language. The data is also well presented without any obvious omissions. Only a couple of points may require clarification.

Are the systematic error bars (black lines e.g. in Figs. 2, 3) implied to be common to entire data sets shown in the figures, or should these be applied independently to each point? This is important to discuss, since these uncertainties are oftentimes of the same magnitude as the effects that are discussed.

It would be helpful to explain whether the theoretical curves (such as in Figs. 2, 4, and some Extended Data figures) have any free parameters that are adjusted to the data.

Finally, the dotted line in Fig. 4(b) is hardly visible.

Referee #4 (Remarks to the Author):

A: first observation of field-linked resonances, with theory providing understanding

B: novelty: a completely new kind of resonance in ultracold molecule-molecule collisions, predicted some time ago but hitherto unseen; significance: opens the way to control molecular interactions in an ultracold gas, typically unavailable by such standard means as Feshbach resonances

C: methodology: requires the production of ultracold molecules, plus microwave technology, both well-known and appropriate in this research group. likewise, the theory appears to be fairly standard, but applied here to good effect.

D: statistics: appropriate

E: conclusions: the whole premise is carefully explained from beginning to end; the data and theory are in encouraging agreement; there are no mysteries as to how the conclusions are arrived at

F: improvements: none

G: references: adequate

H: clarity: the introduction spells out quite clearly the importance of having control of these resonances; figure 1 is adept at explaining the ideas and goals

Author Rebuttals to Initial Comments:

Response to the referee reports

First of all, we thank the referees for reviewing our work and their valuable comments and positive recommendations. We are delighted to read that the referees found our results “excellent” and “beautiful”. We have revised the manuscript according to the referees’ suggestions. In the following response, we address the concerns raised by the referees point-by-point and explain the modifications we have undertaken as a result.

Here we list the main revisions to the manuscript which are described in detail in the following responses:

1. We added “with nuclear spin projections $(m_{i,\text{Na}}, m_{i,\text{K}}) = (3/2, -4)$ ” and “at a magnetic offset field of 72.35 G” to the second sentence in the section “Resonance map”.
2. We now write “common systematic uncertainty” to describe the exemplary systematic error bars in Figs. 2 and 3.
3. We corrected a typo in “Therefore the critical temperature... to about 20% of the Fermi temperature” in the discussion section to “... about 14% of the Fermi temperature”.
4. We added the section “Induced van der Waals interaction” to the Methods.
5. In the Method section “Coupled-channel calculations” we added the second part to the sentence “Hyperfine interactions are not included as these were previously found to have negligible effect [31] when operating at sufficiently large magnetic field [29, 53].” We also added the comment “Note, the coupled-channel calculations have no free parameters.” to the end of this section.
6. We added a citation to Li et al., Nat. Phys. 17, 1144–1148 (2021) to the main text and citations to Karman and Hutson Phys. Rev. A 100, 052704 (2019) and Żuchowski et al., Phys. Rev. A 87, 022706 (2013) to the Methods. We cite now no more than 50 references in the main text.
7. We added missing pre-factors of 2π for the detuning Δ in the captions of Fig. 4 and Extended Data Fig. 3.

RESPONSE TO REPORT FROM REFEREE #1

This paper presents experiment and theory regarding the observation of a predicted new class of “field-linked” states of two weakly bound polar diatomic molecules induced by a resonant microwave field. The existence of such states due to a static electric field was predicted nearly 2 decades ago (Refs. 11,12) and more recently due to an oscillatory electromagnetic field Ref. 14. The field produces a long-range potential that accomplishes two desired outcomes: effective shielding against destructive short-range collisions and the binding of a field-tunable long-range bound tetramer state that makes large changes in the scattering properties of the two molecules. The present paper experimentally measures the properties of such states using the $J = 0$ to 1 rotational transition of the ground state of the NaK molecule induced by microwave radiation and carries out theoretical calculations that account for the measurements and aid in interpreting them. Measured and calculated properties are in semi-quantitative or at least qualitative agreement, indicating that the understanding of the experiment is correct. Since the experiment used identical fermionic molecules, the collisions were three-fold degenerate p-wave ones, with a resonance in one of the p-wave channels; resonances are also expected in s-wave channels of identical bosons or non-identical species. Such field-linked resonances, which are of a generic type to be expected for a range of molecules, give an additional experimental tool by which to manipulate the elastic and inelastic collisional properties of ultracold polar molecules, hopefully leading to new applications in few- and many-body physics and quantum information. There is also a hope, supported by theoretical calculations, that coherent elastic collisions will predominate over lossy inelastic ones by a sufficient amount to enable practical lossless microwave manipulation of molecular quantum gases or lattices.

The results are sound, represent a significant experimental advance demonstrating this new type of microwave-tunable resonance control of ultracold molecular collisions, and are worthy of publishing in Nature.

I found the paper clear in its presentations and graphics. I do not see a need for any significant revisions. I have just a couple points to note that the authors might wish to comment upon.

(A1) We thank the referee for the valuable comments and recommendation of publication.

Looking through the paper again a second time, I could not find which particular hyperfine state of the molecule the authors used. Perhaps this was my oversight, but for the record, the state should be mentioned in the text, even if hyperfine effects turn out to be negligible. Given the great complexity of the hyperfine structure of the field-dressed tetramer potentials and the likelihood of avoided crossing of field-dressed states coupled by inherent potential anisotropies (see Ref. 12 for the static field case), I wonder why hyperfine effects are negligible? The authors do mention that the (previously published) theory suggest that they are. I gather that the mostly isotropic long range shielding repulsion prevents any tunneling to shorter distances where lossy collisions might occur. The losses that are observed here and are predicted then would be due to very-long range couplings among the field-dressed states at distances larger than the shielding repulsion, although it is not clear why hyperfine does not play a role, given the small sub-MHz nuclear spin splitting in a singlet-Sigma molecular ground state (much smaller than the Rabi frequency or detunings used). Any additional enlightenment on this here would be helpful.

(A2) It is true that the Rabi frequency and detunings used are much larger than the hyperfine splittings, meaning that one simultaneously dresses on various hyperfine transitions. We agree with the referee that it is not a priori clear that hyperfine does not affect the dynamics. However, this has been studied in detail in previous theoretical studies including ref. [29], and shown again in our previous study [31], as mentioned in the manuscript.

In short, we are operating at sufficiently high magnetic field so that the hyperfine states become uncoupled from the molecular rotation and act as spectator quantum numbers. This has been shown for many molecular species to occur at magnetic fields around 100 G [29]. We agree with the referee that this is somewhat surprising as by microwave dressing we are mixing $J = 0$ and $J = 1$ rotational states, and for $J = 1$ the Paschen-Back regime is not reached at 100 G in the absence of the microwave field. However, microwave dressing couples $J = 0$ only to a single m_J substate in $J = 1$ ($m_J = 1$ for circular polarization) and precisely because the Rabi frequency is the dominant energy scale, this makes m_J a good quantum number, which helps decoupling the nuclear spin from the molecular rotation.

We appreciate that the manuscript only referred to our previous work [31] to justify the neglect of hyperfine structure. We have now also added a citation to [29] and to Tijs Karman and Jeremy M. Hutson Phys. Rev. A 100, 052704 (2019), which investigated the effect of hyperfine interactions as a function of the magnetic field. Furthermore, we have adapted the manuscript to indicate the used hyperfine state $(m_{i,\text{Na}}, m_{i,\text{K}}) = (3/2, -4)$ and the magnetic field strength of 72.35 G.

The authors use the asymptotic field-dressed van der Waals coefficient for their potentials. I presume this field-dressed repulsive contribution must be large in magnitude compared to the direct attractive van der Waals coefficient, which is known to have a large dipolar contribution for two NaK molecules. They might mention the relative sizes of the direct and field-dressed contributions to justify the neglect of the ordinary term under their conditions.

(A3) Our theory calculations include the direct van-der-Waals coefficient. However, in most of the experimental regime that we covered, the induced van-der-Waals coefficient C_{6i} is much larger than the direct one C_{6d} . We add the following section to the Methods:

Induced van-der-Waals interaction. Reference [15] provides an analytic formula for the induced van-der-Waals interaction

$$U_{\text{vdW}}(\mathbf{r}) = \frac{35C_{6i}}{4r^6} \sin^2 \theta (\cos^2 \theta + 1 - 2 \sin 2\xi \cos^2 \theta \cos 2\varphi - \sin^2 2\xi \sin^2 \theta \cos^2 2\varphi), \quad (1)$$

where the induced C_{6i} coefficient is given by

$$C_{6i} = \frac{d_0^4}{1120\pi^2\epsilon_0^2\Omega(1 + (\Delta/\Omega)^2)^{3/2}}. \quad (2)$$

For $\Omega = 2\pi \times 10$ MHz and zero detuning, $C_{6i} = 1.1 \times 10^{-72}$ kg m⁸/s is orders of magnitude larger than the direct isotropic van-der-Waals coefficient $C_6 = -4.9 \times 10^{-74}$ kg m⁸/s [52]. The induced C_{6i} reduces as detuning increases, and can be comparable to C_6 for $\Delta/\Omega > 2.7$.

RESPONSE TO REPORT FROM REFEREE #2

This is an excellent manuscript reporting the first observation of Feshbach resonances in ultra-cold ground state molecular collisions induced by microwave field. Feshbach resonances have been observed recently in many different systems, however collisions between alkali molecules pose a special problem. At the short range they lead to reaction with nearly unit probability. As such, existence of scattering resonances should be strongly suppressed. In this beautiful experimental work authors build upon ideas of microwave shielding and in particular reference [14]. Microwave radiation couples between two rotational states and generates an adiabatic shallow potential well that supports several bound states. Authors with experimental and theoretical proof that such states indeed exist, with their energy position depending on microwave field ellipticity and detuning. Both parameters can be used in order to tune the position of a quasi-bound state to match the collision energy.

This is a very important achievement, it opens a new window into experiments with polar ultra-cold molecules with tuneable interactions despite the near perfect chemical reactivity that has hindered progress for many years.

(B1) We thank the referee for the favorable comments and positive feedback.

RESPONSE TO REPORT FROM REFEREE #3

The manuscript by X.-Y. Chen et al. reports an experimental investigation of a collisional resonance between ultracold ground-state NaK molecules. These so-called field-linked resonances arise for microwave-dressed molecules when the engineered intermolecular attractive potential well is sufficiently deep and supports (in this case) one or two bound states.

This work pushes ultracold molecular physics to its experimental limits, and reveals a new and quite convincing method to alter molecular interactions on a significant scale. It comes at a time of a rapid progress in quantum control of molecules and is sure to solicit wide-ranging interest from readers. In addition, this method could be applicable more generally and is not expected to be specific to the NaK system. The authors reasonably claim that applications of this control method are not far off.

The paper is well written, using a clear and accessible language. The data is also well presented without any obvious omissions. Only a couple of points may require clarification.

(C1) We thank the referee for the valuable comments and positive recommendation.

Are the systematic error bars (black lines e.g. in Figs. 2, 3) implied to be common to entire data sets shown in the figures, or should these be applied independently to each point? This is important to discuss, since these uncertainties are oftentimes of the same magnitude as the effects that are discussed.

(C2) This systematic error is dominated by uncertainties in the density calibration, which are common to all data points. To clarify this point in the text, we now write “common systematic uncertainty”.

It would be helpful to explain whether the theoretical curves (such as in Figs. 2, 4, and some Extended Data figures) have any free parameters that are adjusted to the data.

(C3) The coupled-channel calculations have no free parameters that are fitted to the data. We now added a comment to the corresponding section in the Methods.

Finally, the dotted line in Fig. 4(b) is hardly visible.

(C4) We have made the dotted line in Fig. 4(b) more visible.

RESPONSE TO REPORT FROM REFEREE #4

A: first observation of field-linked resonances, with theory providing understanding

B: novelty: a completely new kind of resonance in ultracold molecule-molecule collisions, predicted some time ago but hitherto unseen; significance: opens the way to control molecular interactions in an ultracold gas, typically unavailable by such standard means as Feshbach resonances

C: methodology: requires the production of ultracold molecules, plus microwave technology, both well-known and appropriate in this research group. likewise, the theory appears to be fairly standard, but applied here to good effect.

D: statistics: appropriate

E: conclusions: the whole premise is carefully explained from beginning to end; the data and theory are in encouraging agreement; there are no mysteries as to how the conclusions are arrived at

F: improvements: none

G: references: adequate

H: clarity: the introduction spells out quite clearly the importance of having control of these resonances; figure 1 is adept at explaining the ideas and goals

(D1) We thank the referee for the favorable comments and positive recommendation.